# Systematic review and meta-analysis of late auditory evoked potentials as a candidate biomarker in the assessment of tinnitus

Emilie Cardon[1,2]*, Iris Joossen[2], Hanne Vermeersch[2], Laure Jacquemin[1,2],
Griet Mertens[1,2], Olivier M. Vanderveken[1,2], Vedat Topsakal[1,2], Paul Van de Heyning[1,2],
Vincent Van Rompaey[1,2], Annick Gilles[1,2,3]

1 Department of Translational Neuroscience, Faculty of Medicine and Health Science, University of Antwerp, Antwerp, Belgium, 2 University Department of Otorhinolaryngology and Head and Neck Surgery, Antwerp University Hospital, Edegem, Belgium, 3 Department of Education, Health and Social Work, University College Ghent, Ghent, Belgium

☯ These authors contributed equally to this work.
* emilie.cardon@uantwerpen.be

**Data Availability Statement:** The minimal dataset has been made publicly available at the Zenodo public repository (DOI: 10.5281/zenodo.4275525).

## Abstract

Subjective tinnitus, the perception of sound in the absence of any sound source, is routinely assessed using questionnaires. The subjective nature of these tools hampers objective evaluation of tinnitus presence, severity and treatment effects. Late auditory evoked potentials (LAEPs) might be considered as a potential biomarker for assessing tinnitus complaints. Using a multivariate meta-analytic model including data from twenty-one studies, we determined the LAEP components differing systematically between tinnitus patients and controls. Results from this model indicate that amplitude of the P300 component is lower in tinnitus patients (standardized mean difference (SMD) = -0.83, p < 0.01), while latency of this component is abnormally prolonged in this population (SMD = 0.97, p < 0.01). No other investigated LAEP components were found to differ between tinnitus and non-tinnitus subjects. Additional sensitivity analyses regarding differences in experimental conditions confirmed the robustness of these results. Differences in age and hearing levels between the two experimental groups might have a considerable impact on LAEP outcomes and should be carefully considered in future studies. Although we established consistent differences in the P300 component between tinnitus patients and controls, we could not identify any evidence that this component might covary with tinnitus severity. We conclude that out of several commonly assessed LAEP components, only the P300 can be considered as a potential biomarker for subjective tinnitus, although more research is needed to determine its relationship with subjective tinnitus measures. Future trials investigating experimental tinnitus therapies should consider including P300 measurements in the evaluation of treatment effect.

**Funding:** This work was supported by an Applied Biomedical Research grant of the University of Antwerp (FWO T001618 N). The funder provided support in the form of salaries for authors EC, IJ, HV and AG, but did not have any additional role in the study design, data collection and analysis, decision to publish, or preparation of the manuscript. The specific roles of these authors are articulated in the 'author contributions' section.

**Competing interests:** The authors have declared that no competing interests exist.

## Introduction

Tinnitus, commonly defined as the perception of sound in the absence of an external sound source, has a worldwide prevalence of 10–20% [1, 2]. The current paper focuses on subjective tinnitus, in which sound cannot be perceived by the examiner but only by the patient. For a certain segment of patients (10–20%), tinnitus becomes significantly bothersome and interfering with daily life [3, 4]. Tinnitus is often accompanied by nonspecific symptoms such as annoyance, anxiety, depression, hearing problems, hyperacusis, insomnia and concentration difficulties, all of which can add to the burden it places on quality of life [5–8].

Although the tinnitus population is highly heterogeneous in clinical features and underlying pathological mechanisms, one common feature seems to be the maintenance of tinnitus perception by the central nervous system. Central auditory structures are deafferented due to cochlear damage, leading to maladaptive plastic changes in a wide brain network [9, 10]. These widespread maladaptive changes might be reflected in the broad range of nonspecific symptoms that can be observed in chronic tinnitus patients, including cognitive deficits and psychological distress [11, 12]. In addition to a bottom-up mechanism, where chronic understimulation by the auditory periphery ultimately results in abnormal cortical activity, the perception of tinnitus can also be driven by top-down processes, with deficient prefrontal connections failing to suppress ascending signals from thalamic nuclei [13]. The exact contribution of bottom-up versus top-down processes is currently unclear and there is no consensus on a unifying neurophysiological model of tinnitus perception.

Tinnitus patients are systematically evaluated at an outpatient clinic by an ear-nose-throat (ENT) physician, audiologist, and/or psychologist in a multidisciplinary setting [14, 15]. The impact of tinnitus on quality of life and the extent of accompanying symptoms are usually probed using questionnaires, such as the Tinnitus Questionnaire (TQ), the Tinnitus Handicap Inventory (THI) and the Tinnitus Functional Index (TFI) [16–18]. The TFI, especially, has proven to be useful to assess therapy effects while maintaining good validity for discriminative purposes [19, 20]. Questionnaires can be acquired rapidly and with good reliability and validity, and usually require little or no examiner involvement. However, these subjective tools may not be sensitive enough to measure therapy effects in an unambiguous way or to discriminate different patient subtypes. Moreover, there is no widespread consensus on which questionnaire to use.

Attempts to implement more objective measures of tinnitus have proven largely unsuccessful. In this context, psychoacoustic characteristics of the tinnitus sound (i.e. pitch, loudness, and minimum masking level required to render the tinnitus sound inaudible) have received the most attention [21, 22]. However, these characteristics do not seem to correlate with outcomes of tinnitus questionnaires and, thus, are not particularly useful to assess tinnitus severity [23–25]. Additionally, only scarce normative data exist to facilitate the interpretation of psychoacoustic tinnitus characteristics [26]. Some research groups have proposed experimental parameters for objective tinnitus assessment, including cognitive processing speed [27], listening effort [28] and low suppression otoacoustic emission (OAE) amplitudes [29]. However, as these measures have not yet been validated and are challenging to standardize, their use in tinnitus assessment is currently not advised.

Auditory Evoked Potentials (AEPs) might be considered as a potential biomarker in the assessment of tinnitus severity [30, 31]. AEPs can be defined as a series of electrical changes in the peripheral and central nervous system related to auditory processing [32]. AEPs are generally categorized in three classes according to their latency [33]. While auditory brainstem responses (ABRs) and middle latency AEPs indicate the trajectory of sound from the cochlea to the brainstem and subcortical areas, long latency AEPs or late AEPs (LAEPs) reflect perceptual

and cognitive processes resulting from higher brain function in response to auditory events [34]. The earlier responses of the LAEPs (P50, N100, P200, and N200) predominantly reflect aspects of acoustic timing. These are exogenous sensory components that are obligatorily triggered by the presence of a stimulus, but can be affected by subject state. Later LAEP components are mostly endogenous elements, as they mainly reflect neural processes that are task-dependent. For instance, the P300 is a large parietocentral positivity that occurs when a subject detects an informative task-relevant stimulus [35]. This component is often used as a measure of cognitive processing [36, 37] and is thought to reflect the process of updating the neuronal model of the environment when unexpected, but relevant, stimuli are presented [38].

In tinnitus assessment, AEPs might be used as a quick and noninvasive readout of brain activity. AEP outcomes can be meaningfully related to the cognitive and attentional deficits often seen in tinnitus and can serve to broaden the understanding of the underlying tinnitus neurophysiology. For instance, bottom-up tinnitus generation might be reflected in irregularities in earlier LAEP components, whereas deficient top-down processing would be expected to result in alterations of later components. Differences between tinnitus patients and controls have been found both in the early N100 and the late P300 component. However, reporting of LAEPs in tinnitus is highly unsystematic and there is no consensus on which LAEP measures to record.

In order to determine whether LAEPs might be useful as a biomarker of tinnitus presence and severity, it is highly meaningful to explore how they differ in tinnitus patients compared to controls. In this paper, we synthetize relevant data to determine whether LAEPs are different in tinnitus patients and controls, in order to assess appropriateness and efficacy of LAEP use for assessing tinnitus presence, severity and/or treatment effects. We used a multivariate meta-analysis to account for correlations between multiple LAEP components reported within the same groups of subjects. In addition, several categories of bias were investigated to provide guidelines for researchers investigating LAEPs in the tinnitus population.

## Results

### Study selection

A total of 907 records were retrieved from the searched databases (PubMed (MEDLINE): 639; Web of Science: 136; Embase: 127; Cochrane: 5). Hand-searching of the literature revealed three additional records to be screened. The majority of records was excluded following PICOS-based criteria (n = 559). Reviews (n = 33), conference abstracts (n = 23), protocols (n = 5) and foreign language records (n = 78) were also excluded. A summary of the study screening process and reasons for exclusion is provided in the PRISMA flowchart (Fig 1).

Thirty-two papers met all criteria for inclusion in the systematic review. Eleven of these studies were excluded for the final meta-analysis and only considered for narrative review. Detailed characteristics of these studies can be found in section 1 of the S1 Appendix.

### Study characteristics

Twenty-one cross-sectional studies were included for the final meta-analysis. Characteristics of these studies are presented in Table 1. On average, 23.76 tinnitus patients (with the number of included patients ranging from 10 to 55) and 21.14 controls (range 6–51) were included in these studies. Demographic details of included participants in each study can be found in the Section 2 in S1 Appendix. In those papers where mean age of tinnitus patients was reported (n = 19), subjects with tinnitus were on average 42.66 years old (range 23.43–54.8), whereas control subjects had an average mean age of 39.62 years (n = 17, range 21.51–53). The proportion of male participants in the tinnitus group was, on average, 0.61 (ranging from 0.33 to 1), whereas the average proportion of male control subjects was 0.5 (not reported in 2 papers,

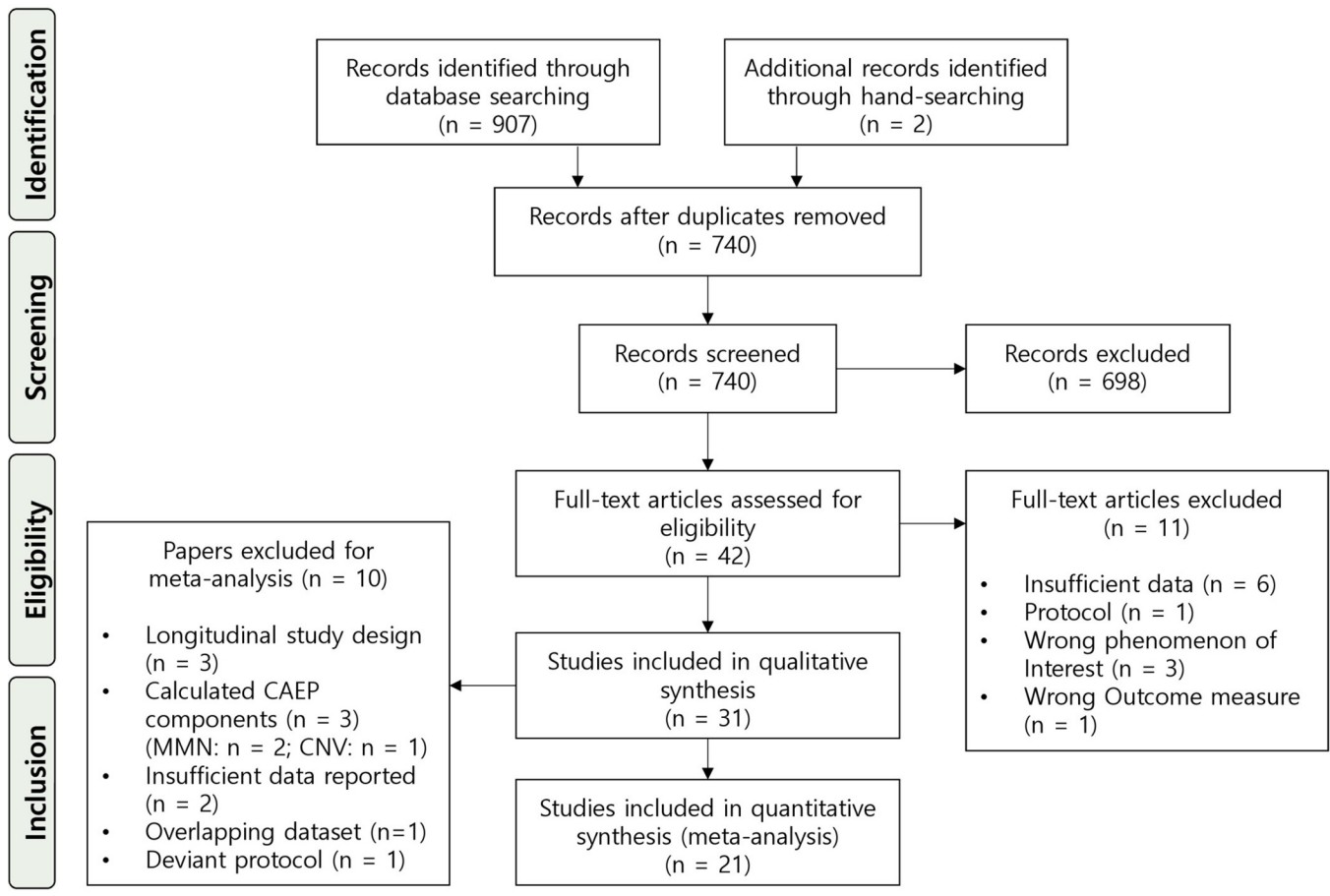

**Fig 1. PRISMA flowchart of the study selection procedure.**

range 0.29–0.87). In the subset of papers where pure tone averages (PTA) for 1, 2 and 4 kHz could be calculated, the average for the tinnitus group was 17.99 dB (n = 9, range 5.33–28.92), while the average PTA for the control subjects was 19.28 (n = 6, range 5.03–28.17). However, the majority of papers did not include sufficient hearing level data and these average hearing thresholds should not be interpreted as representative for all included studies.

Outcome measurements and methods varied across papers. Regarding electrode setup, most papers reported single-channel data (n = 16), with the majority of these including results recorded at the central electrode Cz (n = 10) and some studies only reporting results from the frontal Fz (n = 4) or other electrodes (n = 2). A minority of papers reported data averaged across several different electrodes, either widespread across the whole scalp (n = 1) or clustered in different regions of interest (n = 3). A standard auditory oddball paradigm, consisting of a standard tone presented at 80–85% and a deviant target tone (15–20%), was used in the major-ity of included papers (n = 12). Reported components differed among papers but there was a significant amount of overlap between different studies, necessitating the use of a multivariate model for the final meta-analysis.

## Risk of bias within studies

Selection bias regarding hearing levels was rated as high in three out of the twenty-one included papers (Table 1). In these studies, hearing levels for tinnitus participants were

**Table 1. Characteristics of all studies included in the meta-analysis, including outcome measurements, reported LAEP components and risk of bias.**

| First author (year) | Number of participants | | Outcome measurement | | Component(s) | Risk of bias | | |
|---|---|---|---|---|---|---|---|---|
| | | | Electrode setup | Paradigm | | Selection bias | | Suitability of the LAEP paradigm |
| | Tin | Con | | | | Hearing levels | Other sources | |
| Shiraishi (1991) [39] | 20 | 20 | Fz, Cz, Pz; ground at earlobes | S1-S2 key press task | CNV amplitude, N100 latency and amplitude, P300 latency and amplitude | Low | Low | Low |
| Attias (1993) [40] | 12 | 12 | Fz, Cz, Pz; referred to M1; ground at left forearm | (1) Stimulus counting | N1 latency and amplitude; P200 latency and amplitude; P300 latency and amplitude | Low | Gender: unclear | Low |
| | | | | (2) Standard oddball | | | | |
| | | | | (3) Modified oddball | | | | |
| Attias (1996) [41] | 21 | 21 | Fz, Cz, Pz; referred to M1; ground at M2 | (1) Visual oddball | N100 latency and amplitude; P200 latency and amplitude; N200 latency and amplitude; P300 latency and amplitude | Low | Gender: unclear | Low |
| | | | | (2) Auditory oddball | | | | |
| Jacobson (1996) [42] | 37 | 15 | Cz; referred to A1; ground at Fpz | Oddball | N100 latency and amplitude; P200 latency and amplitude; Negative difference wave | Unclear | Age: high | Low |
| Norena (1999) [30] | 25 | 13 | Fz; referred to A1 and A2 | 200 ms tone bursts | N100 latency; P200 latency; N100-P200 amplitude | Unclear | Gender: high | Low |
| Bilateral tinnitus | 16 | | | | | | | |
| Unilateral tinnitus | 9 | | | | | | | |
| Jacobson (2003) [43] | 32 | 31 | Fz; referred to A1; ground at Fpz | (1) Passive listening | N100 latency and amplitude | High | Gender: high | Low |
| | | | | (2) Oddball | | | | |
| Walpurger (2003) [44] | 10 | 10 | Fz, Cz, Pz; referred to A1 and A2; ground at Fpz | Habituation paradigm using tone pips | N100 latency and amplitude; P200 latency and amplitude; N100-P200 amplitude | Unclear | Low | Low |
| Dornhoffer (2006) [45] | 29 | 35 | Cz; frontally referred; subclavicular ground | Paired rarefaction clicks | P50 amplitude | Unclear | Low | Low |
| Delb (2008) [46] | 41 | 10 | Cz; referred to A1 and A2; ground at Fpz | (1) Passive listening | N100 amplitude | Unclear | Age: high | Low |
| High distress | 15 | | | (2) Modified oddball | | | | |
| Low distress | 26 | | | | | | | |
| Santos Filha (2010) [47] | 30 | 30 | Cz; referred to A1 and A2; ground at Fpz | Oddball | N100 latency; P200 latency; N100-P200 amplitude; P300 latency | Low | Low | Low |
| Gabr (2011) [48] | 40 | 40 | Cz; referred to M1 and M2; ground at Fpz | Oddball | P300 latency | Low | Low | Low |
| Yang (2013) [49] | 20 | 16 | 128-channel cap, reported at Fz; referred to Cz | Oddball | N100 amplitude and latency; P200 amplitude and latency; MMN; LDN | High | Low | High |
| Houdayer (2015) [50] | 17 | 17 | 29-channel cap, reported at electrode displaying greatest ERP | Oddball | N100 amplitude and latency; P200 amplitude and latency; P300 amplitude and latency | Low | Low | Low |
| Hong (2016) [51] | 15 | 15 | 32-channel cap; referred to nose tip; ground at AFz | (1) Passive listening | CNV; N100 amplitude; P200 latency; N200 latency; P300 amplitude and latency | Low | Low | Low |
| | | | | (2) Oddball | | | | |
| Gopal (2017) [52] | 10 | 10 | Fz; referred to A1 and A2; ground at Fpz | Tone bursts | N100 amplitude and latency | Low | Low | Low |

(Continued)

**Table 1.** (Continued)

| First author (year) | Number of participants | | Outcome measurement | | Component(s) | Risk of bias | | |
|---|---|---|---|---|---|---|---|---|
| | | | Electrode setup | Paradigm | | Selection bias | | Suitability of the LAEP paradigm |
| | Tin | Con | | | | Hearing levels | Other sources | |
| **Mannarelli (2017) [53]** | 20 | 20 | 9 central channels; referred to A1 and A2; ground at Fpz | Oddball | N100 amplitude and latency; P300 amplitude and latency | Low | Low | Low |
| **Asadpour (2018) [54]** | 15 | 6 | 32-channel cap; referred to nose tip | (1) Visual oddball | P300 amplitude and latency | Low | Age: high | Low |
| | | | | (2) Auditory oddball | | | | |
| **Durai (2018) [55]** | 16 | 14 | 66-electrode cap, reported at T7 and FC3 | (1) Streaming paradigm | N100(c) amplitude; P200 amplitude | Low | Low | Low |
| | | | | (2) Prediction paradigm | | | | |
| **Morse (2018) [56]** | 13 | 13 | Cz; referred to M1 and M2; ground at Fpz | White noise with silent gap | P50 amplitude and latency; N100 latency; P200 latency; N100-P200 amplitude | Low | Low | Low |
| **Campbell (2019) [57]** | 21 | 45 | 128-channel net, reported at frontal ROI | Gating paradigm | P50 amplitude | Low | Low | Low |
| **Majhi (2019) [58]** | 55 | 51 | Not reported | Not reported | P300 amplitude and latency | High | Low | Unclear |

Tin: Tinnitus group; Con: Control group.

demonstrably worse than in the control group. In a further five papers, this bias was rated as unclear, as these authors did not include sufficient data necessary to judge whether hearing levels were matched between tinnitus and control groups. High bias levels for differences in age between tinnitus and control groups were identified in three papers, while bias regarding gender differences was high in two papers and unclear in an additional two. A second level of bias regarded the suitability of the LAEP paradigm. For one paper, this risk was rated as high, as the majority of tinnitus participants but none of the control subjects had a mild hearing loss at those frequencies that were used for stimulus presentation in the EEG task. In one additional paper, this risk was unclear, as stimulus frequencies used in the EEG task were not provided.

## Synthesis of results

The following LAEP components and characteristics were included in the model: P50 amplitude (n = 3), N100 amplitude (n = 13), N100 latency (n = 11), N100-P200-amplitude (n = 4), P200 amplitude (n = 5), P200 latency (n = 6), P300 amplitude (n = 8), and P300 latency (n = 8). Standardized mean differences (SMDs) between tinnitus patients and controls within each study were calculated for these elements (Fig 2).

The final multivariate model resulted in significant SMDs between tinnitus patients and controls for two out of the eight included LAEP elements (Fig 3). P300 amplitude was shown to be lower in tinnitus patients than controls (SMD = -0.83, $p < 0.01$), while P300 latency was significantly longer in tinnitus patients (SMD = 0.97, $p < 0.01$). P200 amplitude seemed to be slightly higher in tinnitus patients than controls, but this difference did not reach statistical significance (SMD = 0.28, $p = 0.27$). Although many individual papers reported differences between tinnitus and control groups for several of the other included components, especially

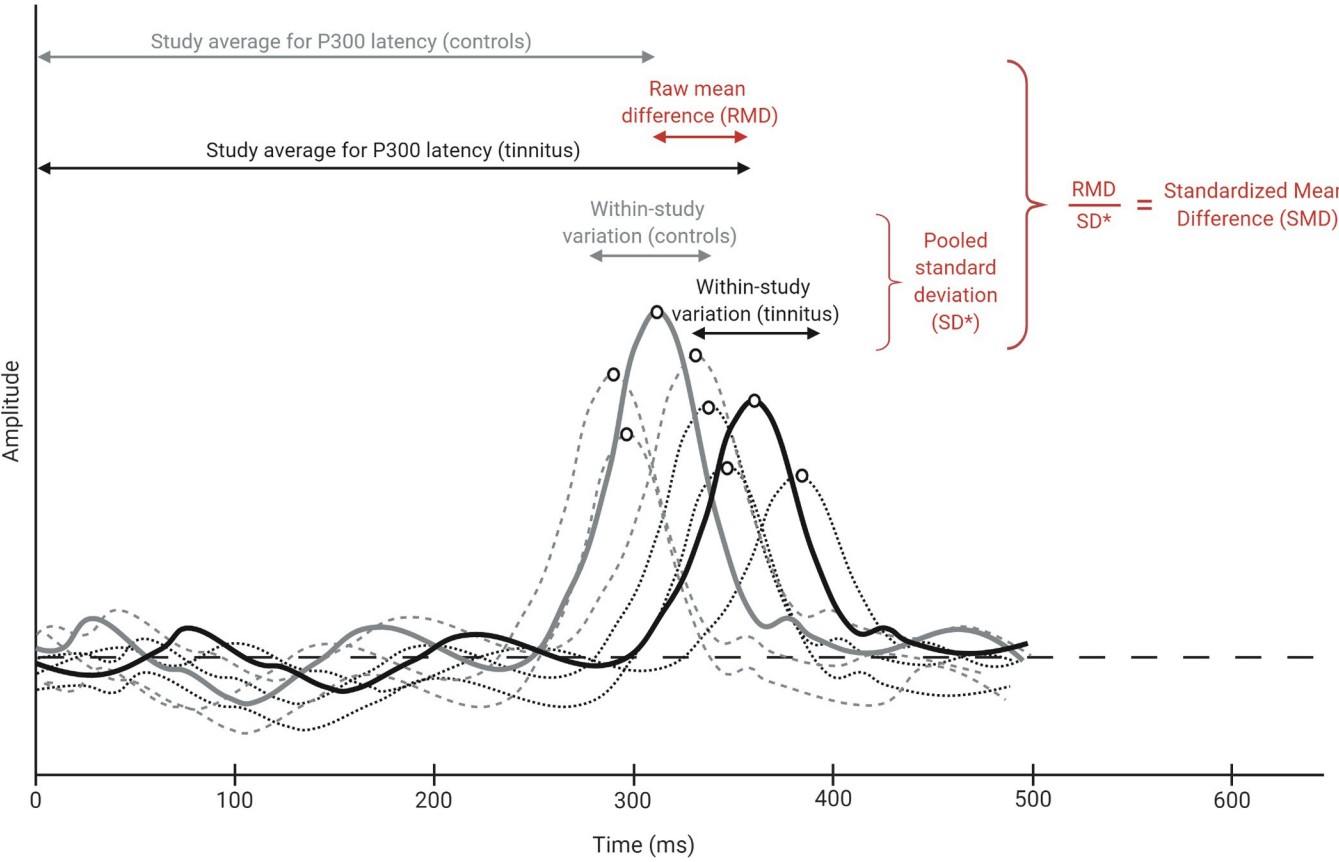

**Fig 2. Calculation of the Standardized Mean Difference (SMD) for P300 latency.** Illustration to clarify the used procedure. Raw mean differences and within-study standard deviations were calculated based on the reported mean values and between-subject variations. SMDs were then calculated and used as input for the multivariate model. Grey lines represent control subjects and black lines represent tinnitus patients. Individual subjects are represented by dotted lines while the average response is presented as a solid line.

amplitude and latency of the N100 component, SMDs for the remaining LAEP elements were close to zero.

Because possible confounding factors might influence different LAEP components in varying ways, the effects of moderators were investigated in separate post hoc analyses for each LAEP element. Differences in gender between tinnitus and control groups did not influence outcomes on either one of the eight LAEP elements. Age difference between tinnitus and non-tinnitus participants was found to be a significant moderator for N100 amplitude ($p < 0.01$) (Section 4 in S1 Appendix). Even after correcting for age difference, no significant difference between tinnitus and control groups was found for this LAEP component. Finally, differences in hearing level could not be accurately assessed as hearing levels were not systematically reported in the included studies.

Possible outliers or influencing studies were also assessed for each component separately (Section 3 in S1 Appendix). Overall, removal of the identified influential papers did not alter the obtained results.

## Risk of bias across studies

Publication bias was investigated using funnel plots and Egger's regression tests for each LAEP element separately in post hoc analyses. Representative funnel plots for P300 amplitude and

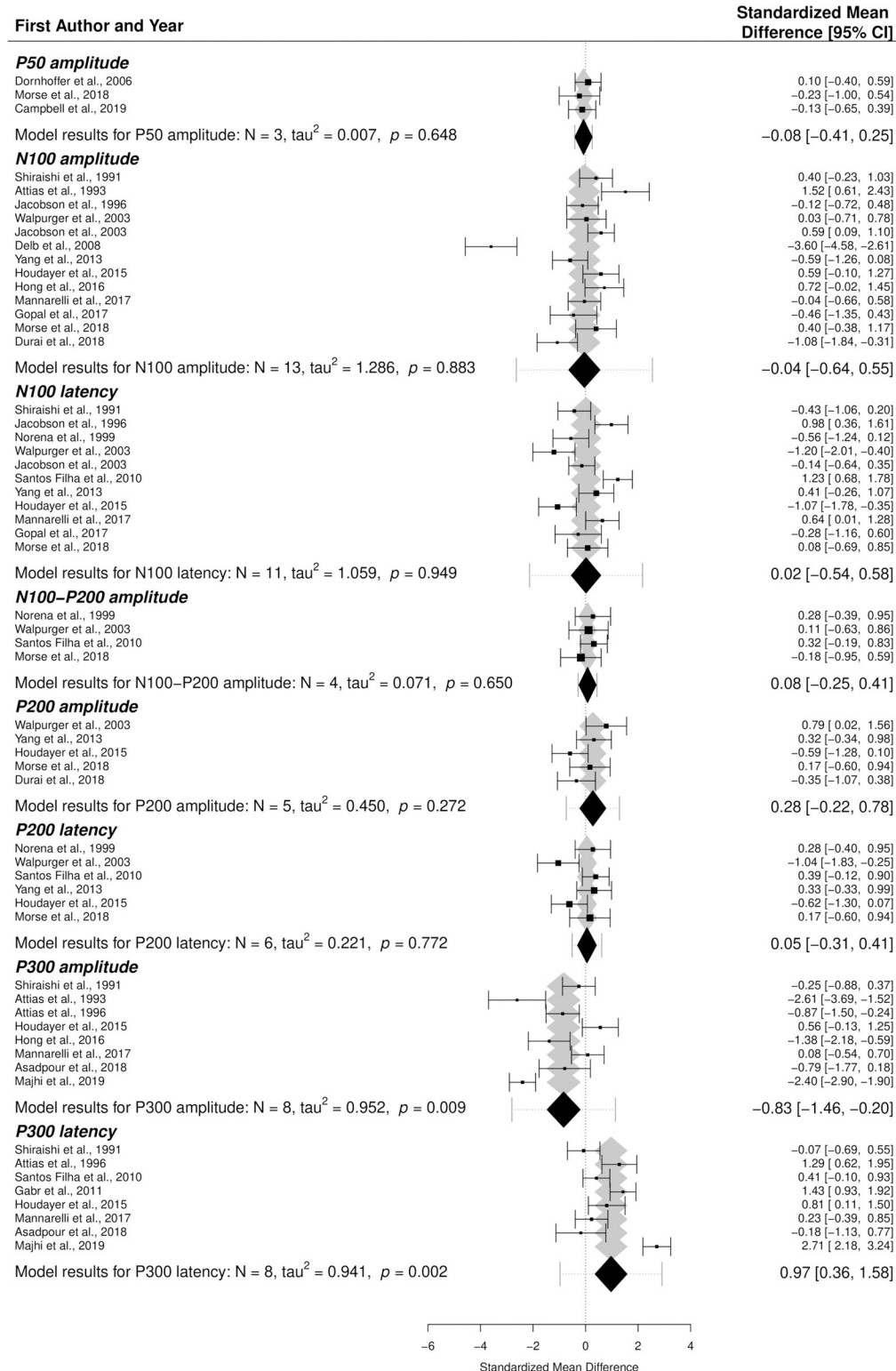

**Fig 3. Forest plot of the primary multivariate analysis.** Results are grouped according to LAEP component. Results from individual papers are presented as standardized mean differences (SMD) ± 95% confidence intervals. Overall results from the primary meta-analytic model are given for each component; diamonds represent SMD with 95% confidence intervals, while error bars correspond to credibility/prediction intervals, defined as the intervals where approximately 95% of the true outcomes are expected to fall.

latency are shown in Fig 4. No evidence for publication bias was found for any of the investigated LAEP elements. However, Egger's regression tests were borderline significant for results regarding both the amplitude ($p = 0.06$) and latency of the N100 component ($p = 0.10$) (Section 5 in S1 Appendix).

## Additional analyses

Additional sensitivity analyses were performed to investigate possible effects of recording electrodes, different LAEP paradigms, inclusion of responses to non-target as opposed to target tones, and inclusion or exclusion of different subgroups of tinnitus patients (for detailed results, see Section 6 in S1 Appendix). Overall, results of these sensitivity analyses did not differ from the primary multivariate model. Only the inclusion of non-target instead of target tones resulted in a subtle difference compared to the primary analysis; next to P300 amplitude and latency, N100 amplitude was also found to differ significantly between tinnitus patients and controls (SMD = -0.55, $p < 0.05$).

## Narrative review of studies excluded for meta-analysis

Details of all studies excluded for meta-analysis can be found in the S1 Table in S1 Appendix. No consistent differences in computed LAEP components (i.e., the mismatch negativity (MMN) and contingent negative variation) could be identified. Other studies excluded for the meta-analysis based on deviating protocols or incomplete data reporting did not present any data conflicting with the outcomes of the primary multivariate model.

Of note, four longitudinal studies assessed LAEP components before and after experimental tinnitus treatments. Only two of these papers recorded the P300 component in their study population [59, 60]. Both of these authors reported a non-significant decrease in P300 latency after treatment, consisting of an auditory training program [59] or transcranial direct current stimulation [60]. Jacquemin et al. reported additional significant decreases in N100, P200 and

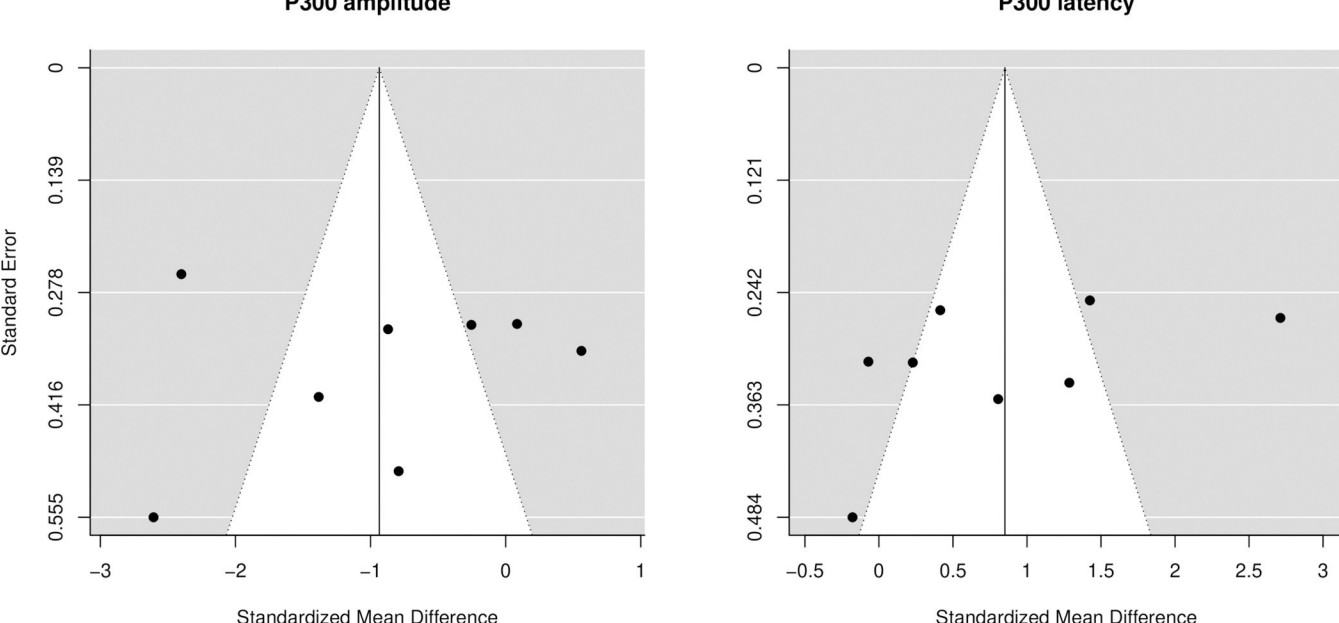

**Fig 4. Forest plots for P300 amplitude (left) and latency (right) show no evidence for publication bias.**

N200 latency and an increase in N200 amplitude. The two remaining papers reported statistically significant differences in MMN amplitudes and N100 latencies after experimental tinnitus treatments [61, 62].

### Correlations between LAEP components and subjective measures

Only a handful of authors reported any correlations between LAEP components and subjective tinnitus severity. The sporadic nature of these reports and the large variability of used questionnaires prevent any quantitative analysis. Regarding the P300 component, one study reported significant positive correlations between its latency and scores on the Hamilton Rating Scale for Depression (HAM-D) and Anxiety (HAM-A), as well as scores on the Mini-Mental State Exam (MMSE) [48]. A different group of authors did not find any significant correlations between P300 amplitude or latency and scores on the THI [53]. No consistent correlations between any of the other LAEP components and subjective tinnitus severity were found.

### Discussion

The current meta-analysis assessed which LAEP components are consistently different between tinnitus patients and controls. Our main finding is a highly specific difference in P300 amplitude and latency between tinnitus and non-tinnitus subjects. Four and three out of the eight included individual papers reported significant differences regarding P300 amplitude and latency, respectively. As amplitude was found to be decreased and latency abnormally prolonged in tinnitus patients, these results point towards a tinnitus-specific impairment in P300-related auditory and/or cognitive processing.

According to the context updating theory, the P300 component occurs when unexpected but relevant stimuli are presented and the neuronal model is revised to better incorporate the incoming data into a pre-existing scheme [63]. As such, the P300 is often used as an index of cognitive efficiency, especially in the context of attention and/or working memory tests [64]. More specifically, higher P300 amplitude and shorter P300 latency are thought to reflect superior cognitive performance. Interestingly, subtle cognitive deficits regarding working memory and executive control of attention have been shown in tinnitus patients [65–67]. Of all included studies in the current meta-analysis, only Gabr et al. determined cognitive performance in their study population [48]. Tinnitus patients were found to score slightly but significantly lower on the Mini-Mental State Exam (MMSE), but remarkably, higher MMSE scores (indicative of a superior cognitive performance) were related to prolonged P300 latencies. Clearly, more studies are necessary to explore any relationship between the P300 component and cognitive efficiency in tinnitus patients, although our results are suggestive of a tinnitus-related impairment in cognitive processing.

There is strong evidence that the P300 component is generated by the simultaneous activation of separate underlying neural structures [31, 35]. As such, a decreased amplitude and prolonged latency might reflect a dysregulation of the synchronized activity of these multiple generators. Combined EEG and fMRI experiments have identified important target-related P300 responses in the temporoparietal cortex but also in limbic structures such as the thalamus and the anterior cingulate cortex (ACC), a node that has been proposed to play a role in conflict detection and response selection [68–70]. Interestingly, there is some evidence suggesting that the ACC is not directly involved in cognition per se, but rather performs the conflict evaluation that serves as the input for top-down attentional processing performed in the dorsolateral prefrontal cortex [71]. The ACC is also part of a frontostriatal circuit involved in the top-down control of tinnitus perception, and grey matter loss in the ACC has been shown to

correspond to tinnitus distress [13, 72]. As part of a limbic processing loop responsible for the valuation of sensory stimuli, the ACC provides input to the prefrontal cortex to send inhibitory signals to subcortical circuits, a process commonly referred to as gain control [73]. In tinnitus, these inhibitory connections are deficient and fail to suppress ascending signals from thalamic nuclei. Thus, in tinnitus patients, the function of the ACC as part of a limbic circuit providing input to top-down prefrontal processes might be compromised, simultaneously resulting in the perception of a tinnitus sound and the observed deficit in the P300 component.

The absence of any consistent differences in earlier LAEP components such as the N100 does not necessarily mean that tinnitus patients do not show any impairment in the earlier stages of auditory processing reflected by these components. Rather, it might be the case that possible deficits at these processing stages do not manifest themselves using a straightforward oddball paradigm. Indeed, some of the studies included here demonstrated differences between tinnitus patients and controls using specific gating or habituation paradigms. More-over, we found that when including responses to non-target instead of only target tones, a significant difference regarding N100 amplitude was identified, with lower amplitudes found in tinnitus patients. This finding suggests that early auditory cortical processing in tinnitus patients might be especially influenced by listening conditions and experimental paradigms. Thus, the results presented here should not be interpreted as a dismissal of tinnitus-related deficits in early auditory processing. Authors exploring possible differences regarding these exogenous LAEP components should carefully select the appropriate paradigm for this research question, and the results that have already been reported should ideally be confirmed in replication studies using similar paradigms.

Next to these considerations regarding paradigm selection, authors conducting studies of LAEP components in tinnitus populations should take special care to use a control group that is well-matched to the experimental group. This issue of matching particularly pertains to hearing level and age. Hearing level importantly influences auditory processing as measured by LAEPs [32]. Notably, many of the studies included here did not report sufficient data on participants' hearing levels. We recommend averaged audiograms or, minimally, pure tone averages of both subject groups and the use of a statistical test to verify the absence of hearing level differences between these groups. Furthermore, within the included studies, age differences were found to influence earlier LAEP components. It should be noted this moderator effect seems to be largely driven by one paper which might be considered an outlier or influential study [46]. However, age has been shown to significantly affect the N100 component, with older adults displaying prolonged latencies and decreased amplitudes compared to younger adults [74]. Thus, in order to exclude possible confounding effects, both age and hearing levels should ideally be similar across subject groups in studies investigating LAEPs in tinnitus. In cases where matching of both groups is not possible, statistical analyses should at least correct for differences between experimental and control groups. A graphical overview of the influence of the factors tinnitus, age and hearing level on LAEPs can be found in Fig 5.

Moreover, careful attention should be payed to environmentally induced factors that are known to influence the auditory evoked response, such as the use of drugs that act on the central nervous system. As individuals in the tinnitus population might be more likely to use medication of this type, such as antidepressants, this could present an additional level of bias in the selected studies [75, 76]. The majority of the discussed authors did not provide any information on pharmacological status of the included participants. We strongly suggest a careful correction for this potentially confounding factor in future studies.

To our knowledge, these results represent the first meta-analysis and systematic review investigating LAEP differences between tinnitus patients and controls. As many of the reported findings have been shown to contradict each other, a quantitative analysis is crucial

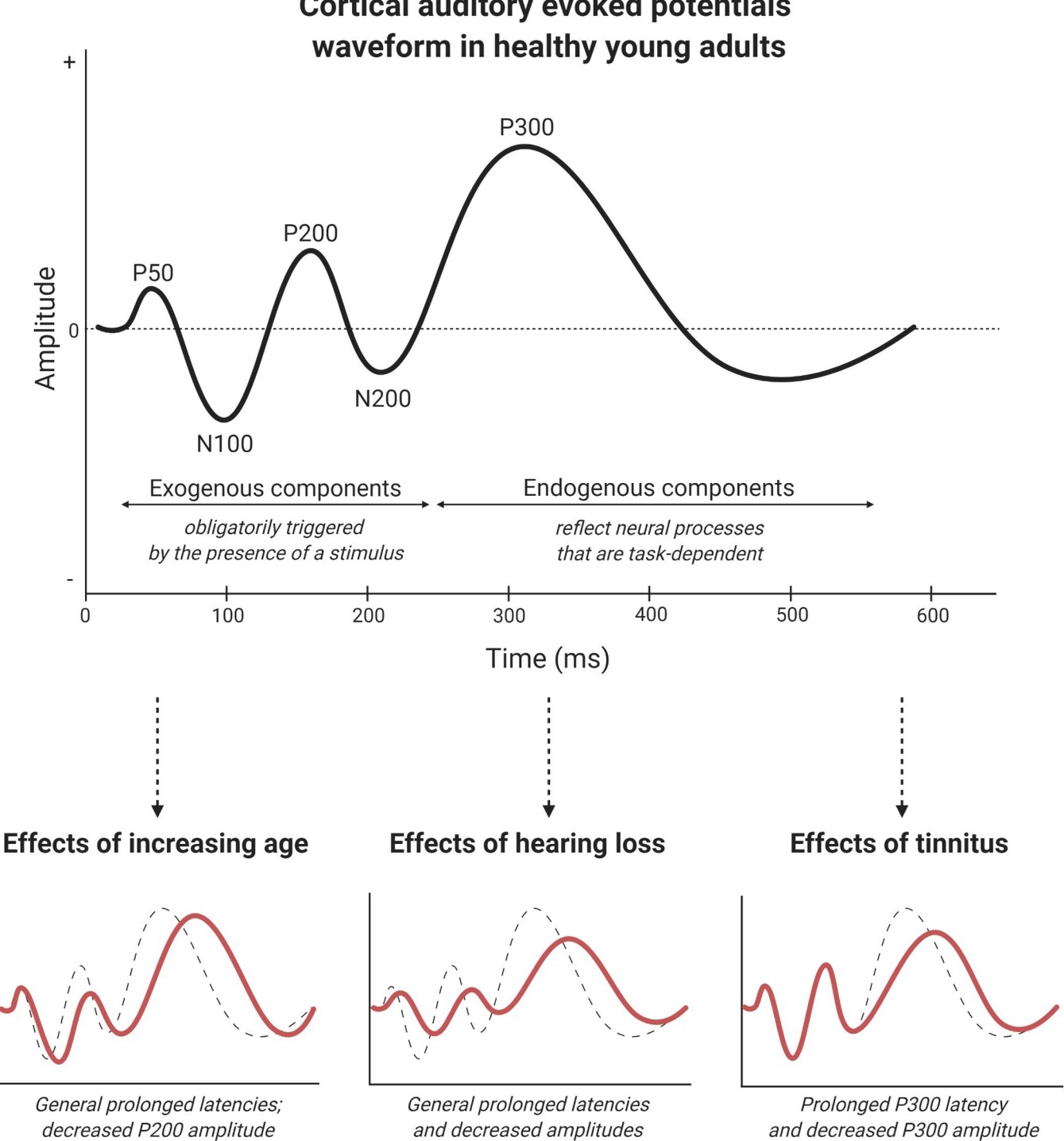

**Fig 5. Graphical overview of the effects of increasing age, hearing loss and tinnitus on the LAEP waveform.** Illustration produced with BioRender.

to explore the robustness of individual results. Although the included data were homogenized as much as possible, the inclusion and pooling of data recorded under subtly different experimental conditions is unavoidable. Differences in experimental paradigms in the included

studies might especially handicap the drawing of robust conclusions, and although additional sensitivity analyses revealed that these and other differences in experimental conditions did not affect our results, the necessary pooling of dissimilar datasets might still be considered suboptimal.

Although we established clear differences in the P300 component between tinnitus patients and controls, we could not detect any conclusive evidence to demonstrate that this component is affected by tinnitus therapy. The few existing studies evaluating tinnitus treatments by use of LAEPs utilized different paradigms and, therefore, could not be evaluated collectively. Only two of these studies recorded the P300 component before and after an experimental treatment, which consisted of an eight-week long auditory training program [59] or transcranial direct current stimulation [60]. Both of these authors reported a small, non-significant decrease in the P300 latency after treatment. The question of whether or not the demonstrated P300-specific deficit in auditory-cognitive processing might be rescued by successful tinnitus therapy should be answered in future randomized controlled trials. Although we demonstrate an average difference in both amplitude and latency of the P300 component between tinnitus patients and controls, in individual studies, this effect might not be apparent due to random sampling of the population. Therefore, we suggest the inclusion of a non-tinnitus control group in longitudinal studies, in order to establish differences in LAEP components at baseline.

Moreover, although the reported differences between tinnitus and non-tinnitus groups were found to be robust, no persuasive evidence was found to argue that subjective tinnitus severity is reflected by LAEPs. This is mainly due to the lack of reporting of correlations between questionnaire scores and LAEP outcomes. Regarding the P300 specifically, one author reported abnormally prolonged latencies for subjects with higher scores on anxiety and depression questionnaires, but correlations between scores on a tinnitus-specific questionnaire (i.e. the THI) and P300 characteristics were reported as non-significant by a different research group. Going forward, we urge future authors to consider reporting these types of correlations, as they could give us the insights necessary to move towards a definitive recommendation of the use of LAEPs as a biomarker of tinnitus severity.

In conclusion, we report a specific LAEP difference between tinnitus patients and controls, as results from our meta-analysis showed that both amplitude and latency of the P300 component are significantly affected in tinnitus patients. These results serve as a confirmation of the crucial involvement of the central nervous system in the maintenance of subjective tinnitus and support the use of LAEPs as a biomarker in future tinnitus research.

## Material and methods

The methods described below follow guidelines based on PRISMA [77], MOOSE [78] and a recent systematic review on systematic review guidelines [79].

### Protocol registration

The protocol for this review was registered at the PROSPERO international prospective register of systematic reviews (PROSPERO ID: CRD42019124690).

### Eligibility criteria

Eligibility criteria regarding study characteristics were based on the PICOS acronym:

- Population: subjective tinnitus patients.

- Phenomenon of Interest: the recording of LAEPs in tinnitus patients.

- Comparator: no constrictions were applied.

- Outcomes: LAEP measures in tinnitus patients compared to controls, or LAEP measures changing longitudinally during treatment.

- Study design: both cross-sectional observational studies and longitudinal experimental randomized controlled trials were included.

Only primary research published in English was considered for this review. There were no restrictions on date of publication.

### Information sources

Databases searched for this review included PubMed (MEDLINE), Embase, Cochrane Library, and Web of Science. The search strategy for PubMed involved the following search string: ('tinnitus' [MeSH] OR 'tinnitus' [tiab] OR 'phantom sound*' [tiab] OR 'ringing' [tiab] OR 'buzzing' [tiab]) AND ('auditory evoked potential*' [MeSH] OR 'event related potential' [MeSH] OR 'auditory evoked potential*' [tiab] OR 'event related potential*' [tiab]). This search string was adapted for all other databases. Additional hand searching of reference lists of relevant articles or reviews was performed.

An initial round of database searching was concluded on January 10[th], 2019. Before the final analyses, databases were searched again to include recent literature. This final round of database searching was completed by November 26[th], 2019.

### Study selection process

A screening of titles and abstracts of the records retrieved from database searches was performed by four investigators in total, such that each record was screened by two independent authors. Records meeting the eligibility criteria and those that could not be excluded based on the title and abstract were subjected to a full-text screening by two independent authors. Disagreements between authors were solved by discussion.

### Data collection process; data items

Data collection was performed using a customized electronic form. This form was piloted on a sample of studies before definitive data collection. Data extraction was performed by two independent authors. Any disagreements between authors were solved by discussion.

The data collection form included fields for study design, characteristics of the participants (number, sex, age, level of hearing loss), type of outcome measures (AEP paradigm and methods, AEP component(s), characteristics (latency, amplitude, scalp distribution), correlations with tinnitus severity, correlations with cognitive measures), statistical method(s) used, and results (differences between tinnitus patients and controls, or longitudinal changes in tinnitus patients). Data were approximated from figures where possible using the WebPlot Digitizer (https://automeris.io/WebPlotDigitizer/). When available, correlation results between LAEP components and tinnitus severity and/or cognitive measures were included in the data collection form.

### Data analysis

Meta-analyses were conducted using the *metafor* package in R (version 3.6.2, © 2019 The R Foundation for Statistical Computing) [80]. Effect sizes were calculated as standardized mean differences between tinnitus groups and control groups. For records where appropriate effect sizes could not be obtained, narrative synthesis was undertaken. As several included papers

reported data on multiple LAEP components within the same group of subjects, sampling errors of these results were expected to be correlated. To account for this correlation, a multivariate model was applied. LAEP components needed to be reported in a minimum of three papers for inclusion in the model; other components were excluded for the final analysis.

In some papers, multiple results for the same LAEP component were reported. These reports concerned results recorded from multiple active electrodes, results obtained in different conditions (e.g. response to target vs. non-target tones), comparisons between multiple groups of tinnitus patients (e.g. 'high distress' vs. 'low distress') and one control group, and/or results obtained using different experimental paradigms (e.g. passive listening vs. auditory oddball paradigm). For the final model, these multiple results were reduced to one singular result according to a fixed set of rules, which are expanded upon in the Section 7 in S1 Appendix. Additional sensitivity analyses were performed to validate the final model and explore possible influences of recording electrode, listening condition, tinnitus subgroup and/or experimental paradigm. Random factors (age, gender, hearing levels) were removed from the multivariate model, as they were found not to affect the overall model outcomes significantly.

Furthermore, separate sensitivity analyses were performed with regard to N100 and P200 amplitudes. Some authors chose to report amplitudes for both components separately, whereas others reported one vertex potential defined as the peak-to-peak amplitude from N100 to P200. The primary model included only data as reported by the authors. Subsequent sensitivity analyses considered separate components only, or combined vertex potentials only (with the vertex potential being calculated as the sum of N100 and P200 amplitudes where possible).

In a multivariate meta-analysis, covariances between the sampling errors of various outcome measures are a necessary addition to the model. However, the information needed to compute these covariances (i.e. the correlations between several outcome measures within one paper) is often not reported. To account for this lack of information, a variance-covariance matrix was constructed based on correlations between different LAEP components in a dataset used in our previously published study on LAEPs in tinnitus patients before and after transcranial direct current stimulation [60].

For those LAEP components included in the multivariate model, post hoc analyses were performed to explore outliers or influential studies. Influence diagnostics were used to visualize influence of individual data points and outlier detection was based on Cook's distance. As outliers and influential cases might reveal important patterns regarding study characteristics that could be acting as potential moderators, the identified influential studies were not removed from the final analysis [81].

To explore the influence of baseline differences between the tinnitus groups and control groups for each component, random factors were added to the post hoc models in a stepwise manner. These factors included differences in age, gender, and hearing levels between the groups. Furthermore, evidence for publication bias was investigated in these post hoc analyses.

## Risk of bias

Overall, three categories of bias were investigated. To explore selection bias, differences in baseline characteristics between tinnitus patients and controls were considered. Special attention was paid to differences in hearing levels, as effects of hearing loss can confound tinnitus-specific effects [82]. Additionally, differences in age and gender between tinnitus and control groups were explored. As a sublevel of reporting bias, publication bias was analyzed by probing potentially unpublished results. Funnel plots were used to explore this level of bias in post hoc analyses, complemented by Egger's regression tests to test for funnel plot asymmetry. Finally, an important additional source of bias might derive from the suitability of the LAEP paradigm.

This bias level was explored by verifying whether stimuli were adequately adapted for hearing impairment, which is present in ca. 90% of tinnitus patients [83].

## Supporting information

**S1 Appendix. Appendix to the main manuscript.**
(DOCX)

**S1 Fig. Age difference is negatively correlated to the standardized mean difference for N100 amplitude (r = -0.10, $p$ < 0.05).**
(TIFF)

**S2 Fig. Funnel plots for N100 amplitude (left) and latency (right) are borderline asymmetric.**
(TIFF)

**S1 File.**
(DOC)

## Author Contributions

**Conceptualization:** Emilie Cardon, Laure Jacquemin, Paul Van de Heyning, Vincent Van Rompaey, Annick Gilles.

**Data curation:** Emilie Cardon, Iris Joossen, Hanne Vermeersch, Annick Gilles.

**Formal analysis:** Emilie Cardon.

**Methodology:** Emilie Cardon.

**Supervision:** Griet Mertens, Olivier M. Vanderveken, Vedat Topsakal, Paul Van de Heyning, Vincent Van Rompaey, Annick Gilles.

**Visualization:** Emilie Cardon.

**Writing – original draft:** Emilie Cardon.

**Writing – review & editing:** Emilie Cardon, Iris Joossen, Hanne Vermeersch, Laure Jacquemin, Griet Mertens, Olivier M. Vanderveken, Vedat Topsakal, Paul Van de Heyning, Vincent Van Rompaey, Annick Gilles.

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
