## [Decision Letter · Decision Letter 0]

6 Oct 2020

PONE-D-20-22709

Systematic review and meta-analysis of late auditory evoked potentials as a candidate biomarker in the assessment of tinnitus

PLOS ONE

Dear Dr. Cardon,

Thank you for submitting your manuscript to PLOS ONE. After careful consideration, we feel that it has merit but does not fully meet PLOS ONE’s publication criteria as it currently stands. Therefore, we invite you to submit a revised version of the manuscript that addresses the points raised during the review process.

We look forward to receiving your revised manuscript.

Kind regards,

Paul Hinckley Delano, Ph.D.

Academic Editor

PLOS ONE

Journal Requirements:

2.In your Data Availability statement, you have not specified where the minimal data set underlying the results described in your manuscript can be found. PLOS defines a study's minimal data set as the underlying data used to reach the conclusions drawn in the manuscript and any additional data required to replicate the reported study findings in their entirety. All PLOS journals require that the minimal data set be made fully available. For more information about our data policy, please see http://journals.plos.org/plosone/s/data-availability.

3.Thank you for stating the following financial disclosure:

 [This work was supported by an Applied Biomedical Research grant of the University of Antwerp (FWO T001618 N). The funders had no role in study design, data collection and analysis, decision to publish, or preparation of the manuscript.].

We note that one or more of the authors is affiliated with the funding organization, indicating the funder may have had some role in the design, data collection, analysis or preparation of your manuscript for publication; in other words, the funder played an indirect role through the participation of the co-authors. If the funding organization did not play a role in the study design, data collection and analysis, decision to publish, or preparation of the manuscript and only provided financial support in the form of authors' salaries and/or research materials, please do the following:

Review your statements relating to the author contributions, and ensure you have specifically and accurately indicated the role(s) that these authors had in your study. These amendments should be made in the online form.

Confirm in your cover letter that you agree with the following statement, and we will change the online submission form on your behalf:

Reviewers' comments:

Reviewer's Responses to Questions

**Comments to the Author**

1. Is the manuscript technically sound, and do the data support the conclusions?

Reviewer #1: Yes

Reviewer #2: Yes

2. Has the statistical analysis been performed appropriately and rigorously? 

Reviewer #1: Yes

Reviewer #2: Yes

3. Have the authors made all data underlying the findings in their manuscript fully available?

Reviewer #1: Yes

Reviewer #2: Yes

4. Is the manuscript presented in an intelligible fashion and written in standard English?

Reviewer #1: Yes

Reviewer #2: Yes

5. Review Comments to the Author

Reviewer #1: This study aim to elucidate the potential role of the auditory evoked response as biomarker in the study of patients with tinnitus. Using a multivariate meta-analytic study, the authors analyze the differences in the Late auditory evoked potentials (LAEPs) between tinnitus patients and controls described in twenty-one articles. The main result was a significantly poorer response of the p300 component in tinnitus patients, manifested by a reduction in amplitude and an increase in latency.

The manuscript is novel, since there is no previously published meta-analysis for the same purpose. The writing is adequate and its reading is fluid to the reader. The authors had a correct control of the Bias and risks, which allows us to think about the development of studies aimed at establishing the p300 component as a biomarker in the study of tinnitus.

In my opinion, there are some minor issues that should be considered for the publication of the manuscript:

Abstract:

L29: I recommend to use the concept potential biomarker

Discussion:

Since it does not appear in fig. 3, I recommend discussing how many of the articles analyzed showed significant differences in latency and / or amplitude of the p300, between the groups

A variable not included in the study is the use of central nervous system depressant medications by tinnitus patients. I recommend discussing it as a potential bias of the study, since it is known that this class of drugs and others affect the auditory evoked response.

Figure:

I recommend improving the resolution of figures 1, 2 and 3 in the final manuscript

Reference

Half of the articles summarized in Table 1 are not in the reference list (shiraishi 1991; Attias 1993, 1996; Jacobson 1996, 2003; Walpurger 2003; Dornhoffer 2006; Yang 2013; Houdayer 2015; Hong 2016, Gopal 2017)

Reviewer #2: This is a systematic review of late auditory potential measurements in tinnitus patients. It is an important piece of work, and I am glad the authors took the effort to produce this review. The outcome is that in tinnitus patients, the P300 component of the auditory response is reduced in amplitudes and delayed in comparison to that in subjects without tinnitus. The authors dealt with the very heterogenous literature in a very thoughtful way.

A consequence is that studies have been dealt with in a heterogeneous way. Therefore

I invite the authors to present the final model of Fig. 3 in some sort of graphical way, clarifying the dependent (various P and N-peaks) and independent (tinnitus, hearing threshold, age) variables

Some additional comments:

L51: This paragraph is a bit odd, as it appears to describe how tinnitus patients are clinically assessed. The first sentence claims that this work is done by an ENT physician or an audiologist. However, in many tinnitus centers the patient is evaluated by both and additionally a psychologist. Here, you only need to briefly summarize the subjective measures (questionnaires etc). I think this paragraph is only intended to contrast the next paragraph, where objective measurements are introduced.

Fig. 2: Is this real data? I think it is just a cartoon to illustrate your computation. Please clarify in the text.

L187: There is a funnel plot for P100 included in the supplemental material. As P300 is the most important outcome, I suggest to include funnel plots of P300 amplitude and delay. I suggest to put this in the main text, rather than in an appendix.

Fig. 3:

- It is confusing that the diamonds in the forest plot also appear as light grey diamonds overlapping with the results of individual studies. I suggest to remove those grey diamonds.

- Please double check the credibility intervals of P300 latency and amplitude. It seems odd that they are much larger than the 95% confidence intervals of the individual studies.

L207: The phrase 'Both of these authors' refers to two multi-author papers. Please correct.

L369: It is unclear to me whether age and hearing threshold were included as co-variates in the multivariate meta-analysis. This is important to do, and age and hearing loss is available for many studies. Please clarify. Please consider to perform a seperate analysis on the subclass of studies where age and hearing loss are both available.

6. PLOS authors have the option to publish the peer review history of their article (what does this mean?). If published, this will include your full peer review and any attached files.

Reviewer #1: No

Reviewer #2: No

---

## [Author Response · Author response to Decision Letter 0]

19 Nov 2020

Please find all responses to the reviewer and editor comments in the attached file 'Response to the reviewers'.

---

## [Editor Report · Decision Letter 1]

26 Nov 2020

Systematic review and meta-analysis of late auditory evoked potentials as a candidate biomarker in the assessment of tinnitus

PONE-D-20-22709R1

Dear Dr. Cardon,

We’re pleased to inform you that your manuscript has been judged scientifically suitable for publication and will be formally accepted for publication once it meets all outstanding technical requirements.

Kind regards,

Paul Hinckley Delano, Ph.D.

Academic Editor

PLOS ONE
---

## [Editor Report · Acceptance letter]

7 Dec 2020

PONE-D-20-22709R1 

Systematic review and meta-analysis of late auditory evoked potentials as a candidate biomarker in the assessment of tinnitus 

Dear Dr. Cardon:

I'm pleased to inform you that your manuscript has been deemed suitable for publication in PLOS ONE. Congratulations! Your manuscript is now with our production department. 

Kind regards, 

on behalf of

Dr. Paul Hinckley Delano 

Academic Editor

PLOS ONE